# Complex Dynamics Analysis of a Discrete Amensalism System with a Cover for the First Species

**Qimei Zhou, Fengde Chen * and Sijia Lin**

School of Mathematics and Statistics, Fuzhou University, Fuzhou 350108, China; 200320015@fzu.edu.cn (Q.Z.); 200320026@fzu.edu.cn (S.L.)
* Correspondence: fdchen@fzu.edu.cn

**Abstract:** Of interest is the dynamics of the discrete-time amensalism model with a cover on the first species. We first obtain the existence and stability of fixed points and the conditions for the permanent coexistence of two species. Then we demonstrate the occurrence of flip bifurcation by using the central manifold theorem and bifurcation theory. A hybrid control strategy is used to control the flip bifurcation and stabilize unstable periodic orbits embedded in the complex attractor. Numerical simulation verifies the feasibility of theoretical analysis and reveals some novel and exciting dynamic phenomena.

**Keywords:** amensalism model; flip bifurcation; chaos control; permanence; global stability

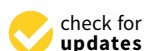



## 1. Introduction

A symbiotic relationship in an ecosystem is a relatively close interrelationship between individuals of different species that can generally increase the fitness of one or both parties. It primarily includes mutualism, commensalism, and amensalism (see [1–8] and references therein). Amensalism refers to the interaction of two species in which only one side is restricted and restrained, while the other side is unaffected. For instance, Dakhama et al. [9] pointed out that Pseudomonas aeruginosa strongly inhibits the growth of green microalgae and cyanobacteria by releasing low molecular weight heat-resistant factors. Penicillin produced by Penicillium can restrain the growth of Gram-positive bacteria [10].

In 2003, Sun [11] firstly established a mathematical model of two species amensalism based on the Lotka–Volterra model, and then Zhu et al. [12] studied the following amensalism model:

$$\begin{cases} \dfrac{\mathrm{d}x}{\mathrm{d}t} = x(\alpha - \beta x - cy), \\ \dfrac{\mathrm{d}y}{\mathrm{d}t} = y(\gamma - \delta y). \end{cases} \tag{1}$$

where $x(t)$ and $y(t)$ represent the densities of the first and second species at time $t$, respectively; $\alpha$ and $\gamma$ stand for the intrinsic growth rates of $x$ and $y$, respectively; $c > 0$ denotes the impact exerted by the second species over the first species. Since then, many scholars have conducted extensive research on the amensalism model based on the model (1). For example, nonlinear functional response [3,13,14], Allee effect [15–19], and a cover (i.e., refuge) [20–22].

Everyone knows that the use of refuge can be broadly defined as including any strategy to reduce the predation rate, such as spatial or temporal refuges, prey aggregation, or reduced prey search activities. Some experimental and theoretical studies have demonstrated that human interference and the interaction between algae and bacteria would make it possible for some algae-killing bacteria to be used for biological control of algal

blooms [9,23,24]. Thus, Xie, Chen, and He [22] investigated a two-species amensalism model with a cover for the first species, as follows:

$$\begin{cases} \dfrac{\mathrm{d}x}{\mathrm{d}t} = \alpha x - \beta x^2 - c(1-k)xy, \\ \dfrac{\mathrm{d}y}{\mathrm{d}t} = \gamma y - \delta y^2, \end{cases} \tag{2}$$

where $k$ denotes a cover provided for the species $x$ and $0 < k < 1$. The remaining parameters are defined as previously. The system admits four possible equilibria $E_0(0,0)$, $E_1(\frac{\alpha}{\beta},0)$, $E_2(0,\frac{\gamma}{\delta})$, and $E_3(x^*,y^*)$. The authors showed that $E_0$ and $E_1$ are unstable, and for the stability property of $E_2$ and $E_3$, the authors obtained the following results.

**Theorem 1.** *(1)　If $0 \le k < \dfrac{\alpha\delta}{\gamma c}$, then $E_2(0,\frac{\gamma}{\delta})$ is globally stable.*

*(2)　If $1 > k > \dfrac{\alpha\delta}{\gamma c}$, then $E_3(x^*,y^*)$ is globally stable.*

　　　Their research shows that the two populations can coexist stably if the cover is large enough. In contrast, if the cover is limited, the first population may be driven to extinction. Here, the dynamic behavior of system (2) seems simple, since from Theorem (1) one could see that the system could not have a bifurcation phenomenon.

　　　In general, discrete-time models described by difference equations are more appropriate and realistic than the continuous-time models when populations have non-overlapping generations. Existing research shows that the discrete-time model not only exhibits more complex dynamic behaviors but also provides more effective numerical simulation results [25–27]. What deserves our attention is that some scholars have studied the discrete-time system with refuge (see [28–34] and references therein). In 2014, Rana et al. [32] considered the impact of the Allee effect and prey refuge on the stability of a discrete predator–prey system, they found that the population remains stable at an intermediate level of refuge parameter, whereas at relatively low and high refuge effects, prey exhibits chaotic oscillation. Santra et al. [33] realized that refuge can stabilize the positive fixed point of the proposed discrete-time model. Recently, a discrete Leslie–Gower model with nonlinear prey harvesting and prey refuge was proposed by Shu and Xie [35]. They researched the existence of flip bifurcation and Neimark–Sacker bifurcation at the internal fixed point. However, up to now, no scholars have investigated the discrete amensalism system with cover.

　　　According to [25,26], the piecewise constant argument method is a better choice for the discretization of continuous models. We firstly consider the following discrete amensalism model:

$$\begin{cases} x_{n+1} = x_n \exp\big(\alpha - \beta x_n - c y_n\big), \\ y_{n+1} = y_n \exp\big(\gamma - \delta y_n\big). \end{cases} \tag{3}$$

　　　Motivated by the above discussions, we then propose the discrete amensalism model with a cover for the first species as follows:

$$\begin{cases} x_{n+1} = x_n \exp\big(\alpha - \beta x_n - c(1-k)y_n\big), \\ y_{n+1} = y_n \exp\big(\gamma - \delta y_n\big), \end{cases} \tag{4}$$

　　　Systems (3) and (4) are discrete versions of continuous systems (1) and (2), respectively. What is the difference between the dynamic behavior of the discrete system (4) and that of the continuous system (2)? What about the dynamics of system (3) and system (4)? The above questions are the primary goals of our research for this paper.

　　　The layout of the paper is as follows: In Section 2, we firstly discuss the existence and stability of fixed points of system (3) and establish a set of sufficient conditions which

ensure the permanence of system (3). Based on system (3), we further incorporate a cover into the first species, that is system (4). In Section 3, we study the existence and stability of its fixed points and give some complete analyses of bifurcation. Moreover, we adopt a hybrid control method to control chaos under the influence of flip bifurcation. Numeric simulations are presented in Section 4 to determine the feasibility of the main results. The paper ends with a brief summary and comparison.

## 2. Dynamics and Bifurcation of System (3)

In this section, we consider the existence and local stability of all possible fixed points.

### 2.1. Analysis of Fixed Points

To obtain the fixed point of (3), we need to solve the following equation:

$$\begin{cases} x = x \exp(\alpha - \beta x - cy), \\ y = y \exp(\gamma - \delta y). \end{cases} \tag{5}$$

Clearly, (3) always has the boundary fixed points $O(0,0)$, $E_1(\frac{\alpha}{\beta}, 0)$, and $E_2(0, \frac{\gamma}{\delta})$. For possible interior fixed point, we have $y = \frac{\gamma}{\delta}$, then substituting it into the first equation of (5), one can know that if $0 < c < \frac{\alpha\delta}{\gamma}$, it has a unique positive root $x_3^* = \frac{\alpha\delta - c\gamma}{\beta\delta}$. The following results can be obtained directly.

**Theorem 2.** *For all parameter values, the following statements are true.*
(1) *It always has three boundary fixed points, which are $O$, $E_1$, and $E_2$.*
(2) *It has only one interior fixed point $E_3^*(x_3^*, \frac{\gamma}{\delta})$ if $0 < c < \frac{\alpha\delta}{\gamma}$.*

The Jacobian matrix of system (3) at a fixed point $E(x, y)$ is

$$J(E) = \begin{pmatrix} (1 - \beta x)M^* & -cxM^* \\ 0 & (1 - \delta y)N^* \end{pmatrix}, \tag{6}$$

where $M^* = \exp(\alpha - \beta x - cy)$, $N^* = \exp(\gamma - \delta y)$. Let $\lambda_1$ and $\lambda_2$ be the two eigenvalues of $J(E)$.

To study the local stability of these fixed points, it is essential to state the following definition, which allows classifying the dynamic behavior of fixed points, see [36] for details.

**Definition 1.** *A fixed point is called*
(1) *a sink if $|\lambda_1| < 1$ and $|\lambda_2| < 1$, and it is locally asymptotically stable;*
(2) *a source if $|\lambda_1| > 1$ and $|\lambda_2| > 1$, and it is unstable;*
(3) *a saddle if $|\lambda_1| > 1$ and $|\lambda_2| < 1$ (or $|\lambda_1| < 1$ and $|\lambda_2| > 1$);*
(4) *non-hyperbolic if either $|\lambda_1| = 1$ or $|\lambda_2| = 1$.*

Now we discuss the types of fixed points and obtain the following theorems.

**Theorem 3.** *$O(0,0)$ is always a source.*

**Proof.** At the boundary fixed point $O(0,0)$, the Jacobian matrix is

$$J(O) = \begin{pmatrix} e^\alpha & 0 \\ 0 & e^\gamma \end{pmatrix},$$

with eigenvalues $\lambda_1 = e^\alpha > 1$ and $\lambda_2 = e^\gamma > 1$. Thus, trivial fixed point $O(0,0)$ is always a source. □

**Theorem 4.** *The local stability of $E_1(\frac{\alpha}{\beta}, 0)$ is briefly described below:*

*(1)  It is a source if and only if $\alpha > 2$;*
*(2)  It is a saddle if and only if $0 < \alpha < 2$;*
*(3)  It is non-hyperbolic if $\alpha = 2$.*

**Proof.** The Jacobian matrix of (6) evaluated at boundary fixed points $E_1(\frac{\alpha}{\beta}, 0)$ would be given by

$$J(E_1) = \begin{pmatrix} 1 - \alpha & -\dfrac{c\alpha}{\beta} \\ 0 & e^\gamma \end{pmatrix}.$$

One can see that the two eigenvalues of $J(E_1)$ are $\lambda_1 = 1 - \alpha < 1$ and $\lambda_2 = e^\gamma > 1$. Hence, $E_1(\frac{\alpha}{\beta}, 0)$ is a source if $\alpha > 2$, a saddle if $0 < \alpha < 2$, and non-hyperbolic if $\alpha = 2$.

For the boundary equilibrium point $E_2(0, \frac{\gamma}{\delta})$, the Jacobian matrix takes the form

$$J(E_2) = \begin{pmatrix} e^{\alpha - \frac{c\gamma}{\delta}} & 0 \\ 0 & 1 - \gamma \end{pmatrix}.$$

The two eigenvalues of $J(E_2)$ are $\lambda_1 = e^{\alpha - \frac{c\gamma}{\delta}} > 0$ and $\lambda_2 = 1 - \gamma < 1$. Since

$$\alpha - \frac{c\gamma}{\delta} \begin{cases} > 0 & \text{if } 0 < c < \frac{\alpha\delta}{\gamma}, \\ = 0 & \text{if } c = \frac{\alpha\delta}{\gamma}, \\ < 0 & \text{if } c > \frac{\alpha\delta}{\gamma}, \end{cases}$$

and

$$1 - \gamma \begin{cases} > -1 & \text{if } 0 < \gamma < 2, \\ = -1 & \text{if } \gamma = 2, \\ < -1 & \text{if } \gamma > 2. \end{cases}$$

Consequently, there are four different topological types for $E_2(0, \frac{\gamma}{\delta})$.  □

**Theorem 5.** *The local stability of $E_2(0, \frac{\gamma}{\delta})$ can be summarized as follows:*

*(1)  It is a sink if and only if $c > \frac{\alpha\delta}{\gamma}$, $0 < \gamma < 2$;*
*(2)  It is a saddle if and only if $0 < c < \frac{\alpha\delta}{\gamma}$, $0 < \gamma < 2$ or $c > \frac{\alpha\delta}{\gamma}, \gamma > 2$;*
*(3)  It is a source if and only if $0 < c < \frac{\alpha\delta}{\gamma}$, $\gamma > 2$;*
*(4)  It is non-hyperbolic if and only if $\gamma = 2$ or $c = \frac{\alpha\delta}{\gamma}$.*

For local stability analysis of the interior fixed point, we obtain the corresponding Jacobian matrix at $E_3^*(x_3^*, \frac{\gamma}{\delta})$ as follows:

$$J(E_3^*) = \begin{pmatrix} 1 - \dfrac{\alpha\delta - c\gamma}{\delta} & \dfrac{-c(\alpha\delta - c\gamma)}{\beta\delta} \\ 0 & 1 - \gamma \end{pmatrix}.$$

It is easy to derive the two eigenvalues of $J(E_3^*)$ are $\lambda_1 = 1 - \frac{\alpha\delta - c\gamma}{\delta} < 1$ and $\lambda_2 = 1 - \gamma < 1$. Note that when $0 < c < \frac{\alpha\delta}{\gamma}$ and $\alpha > 2$, then

$$1 - \frac{\alpha\delta - c\gamma}{\delta} \begin{cases} \in (-1, 1) & \text{if } \frac{(\alpha - 2)\delta}{\gamma} < c < \frac{\alpha\delta}{\gamma}, \\ = -1 & \text{if } c = \frac{(\alpha - 2)\delta}{\gamma}, \\ < -1 & \text{if } 0 < c < \frac{(\alpha - 2)\delta}{\gamma}. \end{cases}$$

Moreover, if $0 < \alpha < 2$ and $0 < c < \frac{\alpha\delta}{\gamma}$, then $1 - \frac{\alpha\delta - c\gamma}{\delta} \in (-1, 1)$. Therefore, the following result can be obtained immediately.

**Theorem 6.** *Suppose that* $0 < c < \frac{\alpha\delta}{\gamma}$, *then the topological classifications of the unique positive fixed point* $E_3^*(x_3^*, \frac{\gamma}{\delta})$ *is given by Table 1.*

**Table 1.** Topological types of the fixed point $E_3^*(x_3^*, \frac{\gamma}{\delta})$.

| Conditions | | | Case |
|---|---|---|---|
| $0 < \alpha \leq 2$ | $0 < c < \frac{\alpha\delta}{\gamma}$ | $0 < \gamma < 2$ | sink |
| | | $\gamma > 2$ | saddle |
| | | $\gamma = 2$ | non-hyperbolic |
| $\alpha > 2$ | $\frac{(\alpha-2)\delta}{\gamma} < c < \frac{\alpha\delta}{\gamma}$ | $0 < \gamma < 2$ | sink |
| | | $\gamma > 2$ | saddle |
| | | $\gamma = 2$ | non-hyperbolic |
| | $0 < c < \frac{(\alpha-2)\delta}{\gamma}$ | $0 < \gamma < 2$ | saddle |
| | | $\gamma > 2$ | source |
| | | $\gamma = 2$ | non-hyperbolic |
| | $c = \frac{(\alpha-2)\delta}{\gamma}$ | $0 < \gamma < 2$ | non-hyperbolic |
| | | $\gamma > 2$ | non-hyperbolic |
| | | $\gamma = 2$ | non-hyperbolic |

We can see from Theorem 6 that flip bifurcation may generate at $E_3^*$, the reason is that the Jacobian matrix has an eigenvalue $-1$. Next, we will establish a set of sufficient conditions which ensure the permanence of system (3).

*2.2. Permanence*

We first introduce the definition of permanence and several useful lemmas [37].

**Definition 2.** *System (3) is said to be permanent if there exist positive constants* $M_i, m_i, (i = 1, 2)$, *such that*

$$m_1 \leq \liminf_{n \to +\infty} x(n) \leq \limsup_{n \to +\infty} x(n) \leq M_1,$$
$$m_2 \leq \liminf_{n \to +\infty} y(n) \leq \limsup_{n \to +\infty} y(n) \leq M_2.$$

**Lemma 1.** *Assume that sequence* $\{u(n)\}$ *satisfies*

$$u(n+1) = u(n) \exp(\alpha - \beta u(n)), n = 1, 2, \dots$$

*where* $\alpha$ *and* $\beta$ *are positive constants and* $u(0) > 0$. *Then*
(1) *If* $\alpha < 2$, *then* $\lim_{n \to \infty} u(n) = \frac{\alpha}{\beta}$.
(2) *If* $\alpha \leq 1$, *then* $u(n) \leq \frac{1}{\beta}, n = 2, 3, \dots$

**Lemma 2.** *Assume that* $x(n)$ *satisfy* $x(n) > 0$ *and*

$$x(n+1) \leq x(n) \exp(a - bx(n)), n \in \mathbb{N},$$

*where a and b are positive constants. Then*

$$\limsup_{n\to+\infty} x(n) \le \frac{\exp(a-1)}{b} := M.$$

**Lemma 3.** *Assume that $x(n)$ satisfy $x(n) > 0$ and*

$$x(n+1) \ge x(n)\exp\left(a - bx(n)\right), n \in \mathbb{N},$$

*where a and b are positive constants. Then*

$$\liminf_{n\to+\infty} x(n) \ge \frac{a}{b}\exp\left(a - bM\right),$$

*where M is given by Lemma 2.*

For the permanence of system (3), the proof process is similar to the literature [15], we only give the following results:

**Theorem 7.** *The second population of system (3) is always persistent.*

**Theorem 8.** *If $\alpha > cM_2$ holds, where $M_2 = \dfrac{\exp(\gamma-1)}{\delta}$, system (3) is always persistent.*

Now we discuss the global stability of $E_3^*(x_3^*, \frac{\gamma}{\delta})$ by developing the analysis technique of Chen [38] and Li and Chen [39].

*2.3. Global Stability of Interior Fixed Point*

From Lemma 1, the following theorems can be easily obtained.

**Theorem 9.** *Assume that $0 < \gamma < 2$ holds, $(x(n), y(n))$ is any positive solution of system (3), then*

$$\lim_{n\to\infty} y(n) = \frac{\gamma}{\delta}.$$

Now we consider the following system

$$x_1(n+1) = x_1(n)\exp\left(\alpha - \frac{c\gamma}{\delta} - \beta x_1(n)\right), \tag{7}$$

whose any positive solution is $x_1(n) = \frac{\alpha\delta - c\gamma}{\beta\delta}$. We obtain the following theorem.

**Theorem 10.** *Suppose that*

$$0 < \gamma < 2,\ 0 < \alpha - \frac{c\gamma}{\delta} < \ln 2 + 1 \tag{8}$$

*holds, $E_3^*(x_3^*, \frac{\gamma}{\delta})$ is globally attractive, that is,*

$$\lim_{n\to+\infty}[x(n) - x_1(n)] = 0,$$

*where $x_1(n)$ is any positive solution of system (7).*

**Proof.** From Theorem 9, for any sufficiently small $\varepsilon > 0$, if there exists an integer $n > N_1$, then

$$y_n > \frac{\gamma}{\delta} - \varepsilon. \tag{9}$$

In order to prove $\lim_{n\to+\infty}[x(n) - x_1(n)] = 0$, we assume that

$$x(n) = x_1(n)\exp[k_1(n)].$$

Then the first equation of (3) is equivalent to

$$
\begin{aligned}
k_1(n+1) &= \ln x(n) + \alpha - \beta x(n) - cy(n) - \ln x_1(n+1) \\
&= k_1(n)\left(1 - \beta x_1(n)\exp[\theta_1(n)k_1(n)]\right) - c\left(y_n - \tfrac{\gamma}{\delta}\right),
\end{aligned}
\tag{10}
$$

where $\theta_1(n) \in [0,1]$. Thus, $x_1(n+1)\exp\left[\theta_1(n)k(n)\right]$ is between $x_1(n)$ and $x(n)$. Now, our main goal is to prove

$$\lim_{n\to+\infty} k_1(n) = 0.$$

Since $x(n)\exp\left[\alpha - c(\tfrac{\gamma}{\delta} + \varepsilon) - \beta x_1(n)\right] \le x(n+1) \le x(n)\exp\left[\alpha - \tfrac{c\gamma}{\delta} - \beta x_1(n)\right]$, from Lemmas 2 and 3, we obtain

$$
\limsup_{n\to+\infty} x(n) \le \frac{\exp\left(\alpha - \tfrac{c\gamma}{\delta} - 1\right)}{\beta} := U_1,
$$

$$
\liminf_{n\to+\infty} x(n) \ge \frac{\alpha - c(\tfrac{\gamma}{\delta} + \varepsilon)}{\beta}\exp\left[\alpha - c(\tfrac{\gamma}{\delta} + \varepsilon) - \beta U_1\right] := V_1.
$$

Moreover, according to (7) and Lemmas 2 and 3, we have

$$
\limsup_{n\to+\infty} x_1(n) \le \frac{\exp\left(\alpha - \tfrac{c\gamma}{\delta} - 1\right)}{\beta} = U_1,
$$

$$
\liminf_{n\to+\infty} x_1(n) \ge \frac{\alpha - \tfrac{c\gamma}{\delta}}{\beta}\exp\left[\alpha - \tfrac{c\gamma}{\delta} - \beta U_1\right] \ge V_1.
$$

Hence, for any sufficiently small $\varepsilon > 0$, there exists an integer $N_2 > N_1$ such that if $n \ge N_2$, then

$$V_1 - \varepsilon \le x(n),\ x_1(n) \le U_1 + \varepsilon,\ n \ge N_2. \tag{11}$$

Assume

$$\lambda_1 = \max\left\{|1 - \beta V_1|, |1 - \beta U_1|\right\}.$$

Then, for any sufficiently small $\varepsilon > 0$, we assume

$$\lambda_{\varepsilon 1} = \max\left\{|1 - \beta(V_1 - \varepsilon)|, |1 - \beta(U_1 + \varepsilon)|\right\}. \tag{12}$$

From (9)–(12), we have

$$
\begin{aligned}
|k_1(n+1)| &\le \max\left\{|1 - \beta(V_1 - \varepsilon)|, |1 - \beta(U_1 + \varepsilon)|\right\}|k_1(n)| + c\varepsilon \\
&= \lambda_{\varepsilon 1} + c\varepsilon,\ n \ge N_2.
\end{aligned}
$$

Then we can get the following equation:

$$|k_1(n)| \le \lambda_{\varepsilon 1}^{n-N_2}|k_1(N_2)| + \frac{1 - \lambda_{\varepsilon 1}^{n-N_2}}{1 - \lambda_{\varepsilon 1}}c\varepsilon,\ n \ge N_2. \tag{13}$$

Since $\lambda_{\varepsilon 1} < 1$ and $\varepsilon$ is sufficiently small, we can get $\lim_{n\to+\infty} k_1(n) = 0$, i.e., $\lim_{n\to+\infty}\left[x(n) - x_1(n)\right] = 0$ set up when $\lambda_1 < 1$. Notice that

$$1 - \beta U_1 < 1 - \beta V_1 < 1,$$

then $\lambda_1 < 1$ is equivalent to

$$1 - \beta U_1 > -1,$$

i.e.,

$$0 < \alpha - \frac{c\gamma}{\delta} < 1 + \ln 2.$$

$\lim_{n \to +\infty} [x(n) - x_1(n)] = 0$ has been proved. Hence, this ends the proof of Theorem 10. □

*2.4. Bifurcation Analysis*

In this subsection, we analyze the existence of flip bifurcation at fixed point $E_1(\frac{\alpha}{\beta}, 0)$, $E_2(0, \frac{\gamma}{\delta})$ and $E_3^*(x_3^*, \frac{\gamma}{\delta})$ by using the central manifold and bifurcation theory [40,41].

2.4.1. Flip Bifurcation at $E_1(\frac{\alpha}{\beta}, 0)$ and $E_2(0, \frac{\gamma}{\delta})$

Firstly, conclusion (3) of Theorem 4 tells us that if $\alpha = 2$ holds, one of the eigenvalues of fixed point $E_1(\frac{\alpha}{\beta}, 0)$ is $-1$, and the other eigenvalues are neither 1 nor $-1$. These conditions imply that all parameters belong to the following collection:

$$F_A = \{(\alpha, \beta) : \alpha = 2, \beta > 0\}.$$

Since the central manifold of system (3) at $E_1(\frac{\alpha}{\beta}, 0)$ is $y = 0$, and it restricted to this central manifold is as follows:

$$x_{n+1} = f(x_n) = x_n \exp(\alpha - \beta x_n).$$

$f'(\frac{\alpha}{\beta}) = -1$ can be calculated quickly. Hence, $E_1(\frac{\alpha}{\beta}, 0)$ can experience flip bifurcation when the parameters are varied in a small range of $F_A$. The image is shown in Figure 1a.

In the same way, suppose that all of the parameters are in the following set, based on conclusion (4) of Theorem 5:

$$F_B = \{(\gamma, \delta) : \gamma = 2, \delta > 0\}.$$

We may conclude that when the parameters change in the small neighborhood of $F_B$, flip bifurcation occurs at the fixed point $E_2(0, \frac{\gamma}{\delta})$. The reason seems to be that at $E_2(0, \frac{\gamma}{\delta})$, the central manifold of system (3) is $x = 0$ and the mode of system (3) constrained to this center manifold is as follows:

$$y_{n+1} = g(y_n) = y_n \exp(\gamma - \delta y_n),$$

then we have $g'(\frac{\gamma}{\delta}) = -1$. The image is presented in Figure 1b.

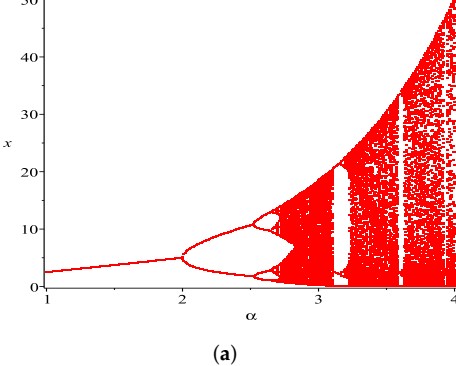

(a)

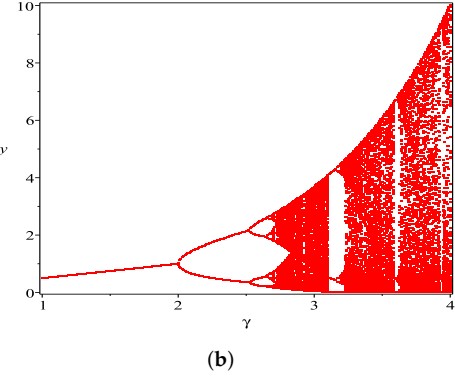

(b)

**Figure 1.** We perceive the parameter values as $\alpha \in [1, 4]$, $\gamma \in [1, 4]$, $\beta = 0.4$, $\delta = 2$ with initial value $(x_0, y_0) = (0.15, 0.2)$. (**a**) Flip bifurcation diagrams of $E_1(\frac{\alpha}{\beta}, 0)$; (**b**) flip bifurcation diagrams of $E_2(0, \frac{\gamma}{\delta})$.

2.4.2. Flip Bifurcation at $E_3^*(x_3^*, \frac{\gamma}{\delta})$

From Theorem 6, system (3) at the positive fixed point $E_3^*(x_3^*, \frac{\gamma}{\delta})$ undergoes flip bifurcation if the condition of $(\alpha, \beta, c_0, \gamma, \delta) \in \Omega_{FB1}$ is satisfied and

$$\Omega_{FB1} := \left\{ (\alpha, \beta, c_0, \gamma, \delta) : c_0 = \frac{(\alpha - 2)\delta}{\gamma}, \ \gamma \neq 2, \ \alpha > 2, \ \beta, \ \gamma, \ \delta > 0 \right\}.$$

Assuming that $\eta$ be a small bifurcation parameter such that $\|\eta\| \ll 1$, then (3) can be expressed by the following two-dimensional map:

$$\begin{pmatrix} x \\ y \end{pmatrix} \rightarrow \begin{pmatrix} x \exp(\alpha - \beta x - (c_0 + \eta)y) \\ y \exp(\gamma - \delta y) \end{pmatrix}. \tag{14}$$

One can see that the mapping (14) has a unique positive fixed point $(x^*, y^*) = \left( \frac{\alpha\delta - (c_0 + \eta)\gamma}{\beta\delta}, \frac{\gamma}{\delta} \right)$. By choosing $u = x - x^*$, $v = y - y^*$, (14) is transferred to

$$\begin{pmatrix} u \\ v \end{pmatrix} \rightarrow \begin{pmatrix} -1 & \dfrac{2\delta(2 - \alpha)}{\beta\gamma} \\ 0 & 1 - \gamma \end{pmatrix} \begin{pmatrix} u \\ v \end{pmatrix} + \begin{pmatrix} f_1(u, v, \eta) \\ g_1(u, v, \eta) \end{pmatrix}, \tag{15}$$

here

$$f_1(u, v, \eta) = z_{110}uv + z_{101}u\eta + z_{020}v^2 + z_{011}v\eta + z_{300}u^3 + z_{201}u^2\eta + z_{120}uv^2$$

$$+ z_{111}uv\eta + z_{030}v^3 + z_{021}v^2\eta + z_{012}v\eta^2 + O((|u| + |v| + |\eta|)^4),$$

$$g_1(u, v, \eta) = (\frac{\delta\gamma}{2} - \delta)v^2 + (-\frac{\gamma\delta^2}{6} + \frac{\delta^2}{2})v^3 + O((|u| + |v| + |\eta|)^4),$$

and

$$z_{110} = \frac{(\alpha - 2)\delta}{\gamma}, \ z_{101} = \frac{\gamma}{\delta}, \ z_{020} = \frac{(\alpha - 2)^2\delta^2}{\beta\gamma^2}, \ z_{011} = \frac{\alpha - 4}{\beta},$$

$$z_{300} = \frac{\beta^2}{6}, \ z_{201} = -\frac{\beta\gamma}{2\delta}, \ z_{120} = -\frac{(\alpha - 2)^2\delta^2}{2\gamma^2}, \ z_{111} = 3 - \alpha,$$

$$z_{030} = -\frac{(\alpha - 2)^3\delta^3}{3\beta\gamma^3}, \ z_{021} = \frac{(2 - \alpha)(\alpha - 6)\delta}{2\beta\gamma}, \ z_{012} = \frac{\gamma}{\beta\delta}.$$

Next, we utilize the following transformation:

$$\begin{pmatrix} u \\ v \end{pmatrix} = \begin{pmatrix} \dfrac{2\delta(\alpha - 2)}{\beta\gamma} & \dfrac{2\delta(\alpha - 2)}{\beta\gamma} \\ 0 & 1 - \gamma \end{pmatrix} \begin{pmatrix} u_1 \\ v_1 \end{pmatrix}.$$

Equation (15) can be written as

$$\begin{pmatrix} u_1 \\ v_1 \end{pmatrix} \rightarrow \begin{pmatrix} -1 & 0 \\ 0 & 1 - \gamma \end{pmatrix} \begin{pmatrix} u_1 \\ v_1 \end{pmatrix} + \begin{pmatrix} f_2(u_1, v_1, \eta) \\ g_2(u_1, v_1, \eta) \end{pmatrix}, \tag{16}$$

where

$$f_2(u_1, v_1, \eta) = s_{110}uv + s_{101}u\eta + s_{020}v^2 + s_{011}v\eta + s_{300}u^3 + s_{201}u^2\eta + s_{120}uv^2$$

$$+ s_{111}uv\eta + s_{030}v^3 + s_{021}v^2\eta + s_{012}v\eta^2 + O((|u| + |v| + |\eta|)^4),$$

$$g_2(u_1, v_1, \eta) = -\frac{\delta}{2}v^2 + \frac{(\gamma - 3)\delta^2}{6(\gamma - 2)}v^3 + O((|u| + |v| + |\eta|)^4),$$

and

$$s_{110} = -\frac{\beta}{2}, \ s_{101} = \frac{\beta\gamma^2}{2(2-\alpha)\delta^2}, \ s_{020} = \frac{(2-\alpha+\gamma)\delta}{2\gamma}, \ s_{011} = \frac{(4-\alpha)\gamma}{2(\alpha-2)\delta},$$

$$s_{300} = \frac{\beta^3\gamma}{12(2-\alpha)\delta}, \ s_{201} = \frac{\beta^2\gamma^2}{4(\alpha-2)\delta^2}, \ s_{120} = \frac{(\alpha-2)\beta\delta}{4\gamma}, \ s_{111} = \frac{(\alpha-3)\beta\gamma}{2(\alpha-2)\delta},$$

$$s_{030} = \frac{\delta^2(-\alpha^2\gamma + \gamma^3 + 2\alpha^2 + 4\alpha\gamma - 3\gamma^2 - 8\alpha - 4\gamma + 8)}{6(2-\gamma)\gamma^2}, \ s_{012} = \frac{\gamma^2}{2(2-\alpha)\delta^2},$$

$$s_{021} = \frac{\alpha-6}{4}, \ u = \frac{2(\alpha-2)\delta}{\beta\gamma}(u_1 + v_1), \ v = (2-\gamma)v_1.$$

According to the central manifold theorem, suppose that an approximate representation of the central manifold $W_1^c(0,0,0)$ is as follows.

$$W_1^c(0,0,0) = \left\{ (u_1, v_1, \eta) : v_1 = \psi(u_1, \eta), \ \psi(0,0) = 0, \ D\psi(0,0) = 0 \right\},$$

where

$$\psi(u_1, \eta) = h_1 u_1^2 + h_2 u_1 \eta + h_3 \eta^2 + O((|u_1| + |\eta|)^3). \tag{17}$$

Applying (16) to both sides of $v_1 = \psi(u_1, \eta)$ synchronously, we have

$$(1-\gamma)v_1 + g_2(u_1, \psi(u_1, \eta), \eta) = h_1 \left[ -u_1 + f_2(u_1, \psi(u_1, \eta), \eta) \right]^2 + h_3\eta^2$$

$$+ h_2 \left[ -u_1 + f_2(u_1, \psi(u_1, \eta), \eta) \right]\eta + O((|u_1| + |\eta|)^3).$$

By comparing the coefficients in the formula above, we obtain $h_1 = h_2 = h_3 = 0$. Therefore, the following expression can be easily computed:

$$F^* : u_1 \to -u_1 + \frac{\gamma}{\delta}u_1\eta + \frac{2\delta^2(\alpha-2)^2}{3\gamma^2}u_1^3 + u_1^2\eta + O((|u_1| + |\eta|)^3).$$

To ensure that the two discriminatory quantities $\tau_1$ and $\tau_2$ are non-zero, where

$$\tau_1 = \left( F^*_{u_1\eta} + \frac{1}{2}F^*_\eta F^*_{u_1 u_1} \right)|_{(u_1,\eta)=(0,0)} = 1 + \frac{\gamma}{\delta},$$

$$\tau_2 = \left( \frac{1}{6}F^*_{u_1 u_1 u_1} + \left( \frac{1}{2}F^*_{u_1 u_1} \right)^2 \right)|_{(u_1,\eta)=(0,0)} = \frac{2\delta^2(\alpha-2)^2}{3\gamma^2}.$$

Therefore, the result below is correct.

**Theorem 11.** *If $(\alpha, \ \beta, \ c_0, \ \gamma, \ \delta) \in \Omega_{FB1}$, then the model (3) undergoes flip bifurcation at $E_3^*(x_3^*, \frac{\gamma}{\delta})$ when the parameter $\eta$ varies in the small neighborhood of origin. Moreover, the period-2 point is attracting due to $\tau_2 > 0$.*

### 3. Dynamics and Bifurcation of System (4)

In this section, we consider the stability of all possible fixed points and bifurcations of system (4). Moreover, we implement a hybrid control strategy of state feedback and parameter perturbation to control the flip bifurcation.

*3.1. Analysis of Fixed Points*

Now, we need to solve the following equation:

$$\begin{cases} x = x\exp(\alpha - \beta x - c(1-k)y), \\ y = y\exp(\gamma - \delta y). \end{cases} \tag{18}$$

By a simple analysis, we obtain the following theorem.

**Theorem 12.** *For any positive parameters,*

(1) *system (4) has three boundary fixed points $O$, $E_1$, and $E_2$, where $O$, $E_1$, and $E_2$ are defined as Theorem 2.*
(2) *if $0 < c < \frac{\alpha\delta}{(1-k)\gamma}$, system (4) has only one positive fixed point $E^*(x^{**}, \frac{\gamma}{\delta})$, where $x^{**} = \frac{\alpha\delta-c(1-k)\gamma}{\beta\delta}$.*

Next, we analyze the local stability of fixed points. The Jacobian matrix of system (4) at the fixed point $Q(x,y)$ is expressed as:

$$J(Q) = \begin{pmatrix} (1-\beta x)M_* & -c(1-k)xM_* \\ 0 & (1-\delta y)N_* \end{pmatrix}, \tag{19}$$

where $M_* = \exp(\alpha - \beta x - c(1-k)y)$, $N_* = \exp(\gamma - \delta y)$.

The local stability analysis of $O$ and $E_1$ are consistent with Theorems 3 and 4. Noting that Jacobian matrix of (19) evaluated at $E_1$ is

$$J(E_1) = \begin{pmatrix} 1-\alpha & -\dfrac{c(1-k)\alpha}{\beta} \\ 0 & e^{\gamma} \end{pmatrix},$$

which is different from the previous one.

At the boundary fixed point $E_2$, the Jacobian matrix takes the form of

$$J(E_2) = \begin{pmatrix} e^{\alpha - \frac{c\gamma(1-k)}{\delta}} & 0 \\ 0 & 1-\gamma \end{pmatrix},$$

with eigenvalues $\lambda_1 = e^{\alpha - \frac{c\gamma(1-k)}{\delta}} > 0$ and $\lambda_2 = 1 - \gamma < 1$. Since

$$\alpha - \frac{c\gamma(1-k)}{\delta} \begin{cases} > 0 & \text{if } 0 < c < \frac{\alpha\delta}{(1-k)\gamma}, \\ = 0 & \text{if } c = \frac{\alpha\delta}{(1-k)\gamma}, \\ < 0 & \text{if } c > \frac{\alpha\delta}{(1-k)\gamma}. \end{cases}$$

One can get the following conclusion.

**Theorem 13.** *The local stability property of $E_2(0, \frac{\gamma}{\delta})$ is discussed as follows:*

(1) *It is a sink if and only if $c > \frac{\alpha\delta}{(1-k)\gamma}, 0 < \gamma < 2$;*
(2) *It is a source if and only if $0 < c < \frac{\alpha\delta}{(1-k)\gamma}, \gamma > 2$;*
(3) *It is non-hyperbolic if and only if $\gamma = 2$ or $c = \frac{\alpha\delta}{(1-k)\gamma}$;*
(4) *It is a saddle for the other values of parameters except for those values in (1)–(3).*

The Jacobian matrix of (19) computed at the positive fixed point $E^*(x^{**}, \frac{\gamma}{\delta})$ is

$$
J(E^*) = \begin{pmatrix} 1 - \dfrac{\alpha\delta - c(1-k)\gamma}{\delta} & -c(1-k)\left(\dfrac{\alpha\delta - c(1-k)\gamma}{\beta\delta}\right) \\ 0 & 1 - \gamma \end{pmatrix}.
$$

The two eigenvalues of $J(E^*)$ are $\lambda_1 = 1 - \frac{\alpha\delta - c(1-k)\gamma}{\delta} < 1$ and $\lambda_2 = 1 - \gamma < 1$. Note that if $0 < c < \frac{\alpha\delta}{(1-k)\gamma}$ and $\alpha > 2$, then

$$
1 - \frac{\alpha\delta - c(1-k)\gamma}{\delta} \begin{cases} \in (-1,1) & \text{if } \frac{(\alpha-2)\delta}{(1-k)\gamma} < c < \frac{\alpha\delta}{(1-k)\gamma}, \\ = -1 & \text{if } c = \frac{(\alpha-2)\delta}{(1-k)\gamma}, \\ < -1 & \text{if } 0 < c < \frac{(\alpha-2)\delta}{(1-k)\gamma}. \end{cases}
$$

If $0 < \alpha < 2$ and $0 < c < \frac{\alpha\delta}{(1-k)\gamma}$ hold, then $1 - \frac{\alpha\delta - c(1-k)\gamma}{\delta} \in (-1,1)$. Therefore, we state directly the following theorem.

**Theorem 14.** *Assuming $0 < c < \frac{\alpha\delta}{(1-k)\gamma}$, the topological classifications of the unique positive fixed point $E^*(x^{**}, \frac{\gamma}{\delta})$ is given by Table 2.*

**Table 2.** Topological types of the fixed point $E^*(x^{**}, \frac{\gamma}{\delta})$.

| | Conditions | | Case |
|---|---|---|---|
| | | $0 < \gamma < 2$ | sink |
| $0 < \alpha \leq 2$ | $0 < c < \frac{\alpha\delta}{(1-k)\gamma}$ | $\gamma > 2$ | saddle |
| | | $\gamma = 2$ | non-hyperbolic |
| | | $0 < \gamma < 2$ | sink |
| | $\frac{(\alpha-2)\delta}{(1-k)\gamma} < c < \frac{\alpha\delta}{(1-k)\gamma}$ | $\gamma > 2$ | saddle |
| | | $\gamma = 2$ | non-hyperbolic |
| | | $0 < \gamma < 2$ | saddle |
| $\alpha > 2$ | $0 < c < \frac{(\alpha-2)\delta}{(1-k)\gamma}$ | $\gamma > 2$ | source |
| | | $\gamma = 2$ | non-hyperbolic |
| | | $0 < \gamma < 2$ | non-hyperbolic |
| | $c = \frac{(\alpha-2)\delta}{(1-k)\gamma}$ | $\gamma > 2$ | non-hyperbolic |
| | | $\gamma = 2$ | is non-hyperbolic |

It can be observed from Table 2 that when the Jacobian matrix has an eigenvalue of $-1$, system (4) may experience flip bifurcation at $E^*(x^{**}, \frac{\gamma}{\delta})$. Next, we will study the permanence of system (4).

*3.2. Permanence*

According to Lemmas 2 and 3, the second equation of (4) satisfies

$$
m_2 \leq \liminf_{n \to +\infty} y(n) \leq \limsup_{n \to +\infty} y(n) \leq M_2,
$$

where

$$
m_2 = \frac{\gamma \exp(\gamma - \delta M_2)}{\delta}, \quad M_2 = \frac{\exp(\gamma - 1)}{\delta}. \tag{20}
$$

Thus, we establish the following permanence results for system (4).

**Theorem 15.** *The second population of system (4) is always persistent.*

**Theorem 16.** *If $\alpha > c(1-k)M_2$ holds, where $M_2$ is defined by Equation (20), then system (4) is always persistent.*

**Proof.** From the condition of Theorem 16, for a sufficiently small positive number $\varepsilon > 0$ with $\alpha > c(1-k)(M_2 + \varepsilon)$. According to Theorem 15, for the above $\varepsilon > 0$, there exists an integer $N_3 > 0$ such that if $n > N_3$, then

$$m_2 - \varepsilon < y(n) < M_2 + \varepsilon. \tag{21}$$

By combining Equation (21) with the first equation of (4), if $n > N_3$, we have

$$x(n+1) \leq x(n)\exp\left\{\alpha - c(1-k)(m_2 - \varepsilon) - \beta x(n)\right\}. \tag{22}$$

Applying Lemma 2 to (22) leads to

$$\limsup_{n \to +\infty} x(n) \leq \frac{1}{\beta}\exp\left\{\alpha - c(1-k)(m_2 - \varepsilon) - 1\right\}. \tag{23}$$

Let $\varepsilon \to 0$, (23) becomes

$$\limsup_{n \to +\infty} x(n) \leq \frac{1}{\beta}\exp\left\{\alpha - c(1-k)m_2 - 1\right\} := M_1. \tag{24}$$

Similarly, by combining Equation (21) with the first equation of (4), if $n > N_3$, we have

$$x(n+1) \geq x(n)\exp\left\{\alpha - c(1-k)(M_2 + \varepsilon) - \beta x(n)\right\}. \tag{25}$$

Applying Lemma 3 to (25) leads to

$$\liminf_{n \to +\infty} x(n) \geq \frac{\alpha - c(1-k)(M_2 + \varepsilon)}{\beta}\exp\left\{\alpha - c(1-k)(M_2 + \varepsilon) - \beta M_1\right\}. \tag{26}$$

Let $\varepsilon \to 0$, (26) becomes

$$\liminf_{n \to +\infty} x(n) \geq \frac{\alpha - c(1-k)M_2}{\beta}\exp\left\{\alpha - c(1-k)M_2 - \beta M_1\right\} := m_1. \tag{27}$$

To sum up, (24), (27), and Theorem 15 show that system (4) is permanent. □

*3.3. Global Stability of Interior Fixed Point*

In this subsection, using the method of iteration scheme, we consider the global stability of interior fixed point $E^*(x^{**}, \frac{\gamma}{\delta})$. The following theorems can be directly obtained.

**Theorem 17.** *Assume that $0 < \gamma < 2$, $(x(n), y(n))$ is any positive solution of system (4), then*

$$\lim_{n \to \infty} y(n) = \frac{\gamma}{\delta}.$$

Next, we regard the following system

$$x_2(n+1) = x_2(n)\exp\left(\alpha - \frac{c(1-k)\gamma}{\delta} - \beta x_2(n)\right), \tag{28}$$

whose any positive solution is $x_2(n) = \frac{\alpha\delta - c(1-k)\gamma}{\beta\delta}$. We obtain the following theorem.

**Theorem 18.** *Suppose that*

$$0 < \gamma < 2, \ 0 < \alpha - \frac{c(1-k)\gamma}{\delta} < \ln 2 + 1, \tag{29}$$

$E^*(x^{**}, \frac{\gamma}{\delta})$ *is globally attractive, that is,*

$$\lim_{n \to +\infty} [x(n) - x_2(n)] = 0,$$

*where $x_2(n)$ is any positive solution of system (28).*

The proof process is similar to Theorem 10, so we omit it.

System (4) experiences the flip bifurcation at fixed points $E_1$ and $E_2$. See Section 2.4.1 for details. Next, we analyze the flip bifurcation at positive fixed point $E^*(x^{**}, \frac{\gamma}{\delta})$ of system (4) by using central manifold theorem and bifurcation theory [40,41].

*3.4. Bifurcation Analysis*

The two eigenvalues of $J(E^*)$ are $\lambda_1 = -1, |\lambda_2| \neq 1$, which leads to $\alpha > 2, c^* = \frac{(\alpha-2)\delta}{(1-k)\gamma}$ and $\gamma \neq 2$. It can be converted into the following set:

$$\Omega_{FB2} := \left\{ (\alpha, \ \beta, \ c^*, \ \gamma, \ \delta) : c^* = \frac{(\alpha-2)\delta}{(1-k)\gamma}, \ \gamma \neq 2, \ \alpha > 2, \ 0 < k < 1, \ \beta, \ \gamma, \ \delta > 0 \right\}.$$

We regard $\xi$ as a new dependent variable parameter, where $\xi = c - c^*$, then (4) can be written as:

$$\begin{pmatrix} x \\ y \end{pmatrix} \to \begin{pmatrix} x \exp\left(\alpha - \beta x - (c^* + \xi)(1-k)y\right) \\ y \exp\left(\gamma - \delta y\right) \end{pmatrix}. \tag{30}$$

Next, choosing $u_2 = x - \frac{\alpha\delta - (c^* + \xi)(1-k)\gamma}{\beta\gamma}$, $v_2 = y - \frac{\gamma}{\delta}$ and expanding the Taylor series around $(u_2, v_2, \xi) = (0, 0, 0)$ to the third order, then (30) becomes

$$\begin{pmatrix} u_2 \\ v_2 \end{pmatrix} \to \begin{pmatrix} -1 & \dfrac{2\delta(2-\alpha)}{\beta\gamma} \\ 0 & 1 - \gamma \end{pmatrix} \begin{pmatrix} u_2 \\ v_2 \end{pmatrix} + \begin{pmatrix} f_3(u_2, v_2, \xi) \\ g_3(u_2, v_2, \xi) \end{pmatrix}, \tag{31}$$

where

$$f_3(u_2, v_2, \xi) = z_{110}u_2v_2 + a_{101}u_2\xi + z_{020}v_2^2 + a_{011}v_2\xi + z_{300}u_2^3 + a_{201}u_2^2\xi + z_{120}u_2v_2^2$$
$$+ a_{111}u_2v_2\xi + z_{030}v_2^3 + a_{021}v_2^2\xi + a_{012}v_2\xi^2 + O\left((|u_2| + |v_2| + |\xi|)^4\right),$$
$$g_3(u_2, v_2, \xi) = (\frac{\delta\gamma}{2} - \delta)v_2^2 + (-\frac{\gamma\delta^2}{6} + \frac{\delta^2}{2})v_2^3 + O\left((|u_2| + |v_2| + |\xi|)^4\right),$$

and

$$a_{101} = (1-k)z_{101}, \ a_{011} = (1-k)z_{011}, \ a_{201} = (1-k)z_{201},$$
$$a_{111} = (1-k)z_{111}, \ a_{021} = (1-k)z_{021}, \ a_{012} = (1-k)^2 z_{012}.$$

By using the transformation of $(u_2, v_2) = T(u_3, v_3)$, where

$$T = \begin{pmatrix} \dfrac{2\delta(\alpha-2)}{\beta\gamma} & \dfrac{2\delta(\alpha-2)}{\beta\gamma} \\ 0 & 1 - \gamma \end{pmatrix}.$$

Equation (31) can be represented as

$$
\begin{pmatrix} u_3 \\ v_3 \end{pmatrix} \rightarrow \begin{pmatrix} -1 & 0 \\ 0 & 1-\gamma \end{pmatrix} \begin{pmatrix} u_3 \\ v_3 \end{pmatrix} + \begin{pmatrix} f_4(u_3, v_3, \xi) \\ g_4(u_3, v_3, \xi) \end{pmatrix},
$$

where

$$
\begin{aligned}
f_4(u_3, v_3, \xi) = {} & s_{110} u_2 v_2 + b_{101} u_2 \xi + s_{020} v_2^2 + b_{011} v_2 \xi + s_{300} u_2^3 + b_{201} u_2^2 \xi + s_{120} u_2 v_2^2 \\
& + b_{111} u_2 v_2 \xi + s_{030} v_2^3 + b_{021} v_2^2 \xi + b_{012} v_2 \xi^2 + O\big((|u_2| + |v_2| + |\xi|)^4\big),
\end{aligned}
$$

$$
g_4(u_3, v_3, \xi) = -\frac{\delta}{2} v_2^2 + \frac{(\gamma - 3)\delta^2}{6(\gamma - 2)} v_2^3 + O\big((|u_2| + |v_2| + |\xi|)^4\big),
$$

and

$$
b_{101} = (1-k)s_{101}, \ b_{011} = (1-k)s_{011}, \ b_{201} = (1-k)s_{201},
$$

$$
b_{111} = (1-k)s_{111}, \ b_{021} = (1-k)s_{021}, \ b_{012} = (1-k)^2 s_{012}.
$$

In addition, we have $u_2 = \frac{2(\alpha-2)\delta}{\beta\gamma}(u_3 + v_3)$, $v_2 = (2-\gamma)v_3$. Utilizing the center manifold theorem, there exists a center manifold $W_2^c(0,0,0)$, which can be approximately expressed as

$$
W_2^c(0,0,0) = \left\{ (u_3, v_3, \xi) : v_3 = k_1 u_3^2 + k_2 u_3 \xi + k_3 \xi^2 + O\big((|u_3| + |\xi|)^3\big) \right\},
$$

for $u_3$ and $\xi$ sufficiently small. By a simple coefficient comparison, $k_1 = k_2 = k_3 = 0$ can be obtained quickly. Therefore, the following expression can be easily evaluated:

$$
G^* : u_3 \rightarrow -u_3 + \frac{\gamma}{\delta} u_3 \xi + \frac{2\delta^2(\alpha-2)^2}{3\gamma^2} u_3^3 + (1-k)u_3^2 \xi + O\big((|u_3| + |\xi|)^4\big).
$$

Consequently, the map $G^*$ undergoes a flip bifurcation if two discriminatory quantities are satisfied $\varpi_1$ and $\varpi_2$ are non-zero, where

$$
\varpi_1 = \big(G^*_{u_3\eta} + \tfrac{1}{2} G^*_\eta G^*_{u_3 u_3}\big)\big|_{(u_3,\xi)=(0,0)} = 1 + \frac{\gamma}{\delta} - k,
$$

$$
\varpi_2 = \big(\tfrac{1}{6} G^*_{u_3 u_3 u_3} + (\tfrac{1}{2} G^*_{u_3 u_3})^2\big)\big|_{(u_3,\xi)=(0,0)} = \frac{2\delta^2(\alpha-2)^2}{3\gamma^2}.
$$

Note that $(\alpha, \beta, c^*, \gamma, \delta) \in \Omega_{FB2}$, it can be testified that $\varpi_1 > 0$ and $\varpi_2 > 0$. Therefore, we have the following theorem.

**Theorem 19.** *If $(\alpha, \beta, c^*, \gamma, \delta) \in \Omega_{FB2}$, then system (4) experiences flip bifurcation at $E^*(x^{**}, \frac{\gamma}{\delta})$ when the parameter $\xi$ varies in the small neighborhood of origin. Since $\varpi_2 > 0$, the period-2 points that bifurcate from $E^*(x^{**}, \frac{\gamma}{\delta})$ are stable.*

### 3.5. Chaos Control

The hybrid control method [42] is a common strategy to control the bifurcation and chaotic behavior of the discrete-time model. The main contribution of this subsection is to control the flip bifurcation by using a hybrid control method.

For the application of the hybrid control method, system (4) can be written in the following form:

$$
\begin{cases} x_{n+1} = \rho x_n \exp\big(\alpha - \beta x_n - c(1-k)y_n\big) + (1-\rho)x_n, \\ y_{n+1} = \rho y_n \exp\big(\gamma - \delta y_n\big) + (1-\rho)y_n, \end{cases} \tag{32}
$$

where $0 < \rho < 1$. The Jacobian matrix of (32) is evaluated at the positive fixed point $E^*(x^{**}, \frac{\gamma}{\delta})$ as follows:

$$J(E^*) = \begin{pmatrix} 1 - \dfrac{\rho(\alpha\delta - c(1-k)\gamma)}{\delta} & -\dfrac{\rho c(1-k)(\alpha\delta - c(1-k)\gamma)}{\beta\delta} \\ 0 & 1 - \rho\gamma \end{pmatrix}. \qquad (33)$$

One can see that the eigenvalues of $J(E^*)$ are $\lambda_1 = 1 - \dfrac{\rho(\alpha\delta - c(1-k)\gamma)}{\delta} < 1$ and $\lambda_2 = 1 - \rho\gamma < 1$. Hence we obtain the following theorem:

**Theorem 20.** *The positive fixed point $(x^{**}, \frac{\gamma}{\delta})$ of the controlled system (33) is locally asymptotically stable if and only if*

$$0 < \rho < \min\left\{\frac{2}{\gamma}, \frac{2\delta}{\alpha\delta - c(1-k)\gamma}, 1\right\}.$$

## 4. Numerical Examples and Discussions

In this section, our purpose is to demonstrate the viability of the above main results. We run numerical simulations of systems (3) and (4), including bifurcation diagrams, phase diagrams, maximum Lyapunov exponents, etc.

**Example 1.** *The change in the dynamical behavior of system (3) is illustrated by varying the parameters $\alpha$ (the intrinsic growth rate of x) and $\gamma$ (the intrinsic growth rate of y). The parameter values of system (3) are as follows:*

(a) *Varying c in range $0 < c < 3$, and fixing $\alpha = 1.5, \beta = 0.4, \gamma = 1.5, \delta = 2$;*
(b) *Varying c in range $0 < c < 2$, and fixing $\alpha = 1.5, \beta = 0.4, \gamma = 2.5, \delta = 2$;*
(c) *Varying c in range $0 < c < 6$, and fixing $\alpha = 4, \beta = 0.4, \gamma = 1.5, \delta = 2$;*
(d) *Varying c in range $0 < c < 4$, and fixing $\alpha = 4, \beta = 0.4, \gamma = 2.5, \delta = 2$.*

*From Figure 2a, one can see that system (3) has only one positive fixed point $E_3^*(-1.875c + 3.75, 0.75)$ when $0 < c < 2$, and this point is stable. If $c > 2$, there has no positive fixed point. We see from Figure 2b that there has a unique positive fixed point $E_3^*(-3.125c + 3.75, 0.75)$ when $0 < c < 1.2$, but it is unstable. If $c > 1.2$ holds, system (3) does not have positive fixed point. The above results support Theorems 2 and 6. We also know that flip bifurcation occurs at $E_3^*(5, 0.75)$ for $c_0 = 2.667$ (see Figure 2c for the bifurcation diagram). Moreover, $\tau_1 = 1.75, \tau_2 = 4.741$ can be easily calculated, and the period-2 point is attracting due to $\tau_2 > 0$. Note that if $c > 5.333$, there has no positive fixed point. From Figure 2d, if $c > 3.2$, system (3) has no positive fixed point, and $E_3^*(-3.125c + 10, 1.25)$ is unstable when $c \in (0, 3.2)$. According to the numerical simulation above, we believe that the internal growth rate of the second population plays a decisive role in the stability of system (3). Furthermore, setting the parameter values as $\alpha = 4, \beta = 0.4, c = 4, \gamma = 1.5, \delta = 2$ with the same initial value $(x_0, y_0) = (0.15, 0.2)$. From the conditions of Theorems 10 and 18, the only one interior fixed point $E_3^*(2.5, 0.75)$ and $E^{**}(4.75, 0.75)$ are globally stable. The image is shown in Figure 3.*

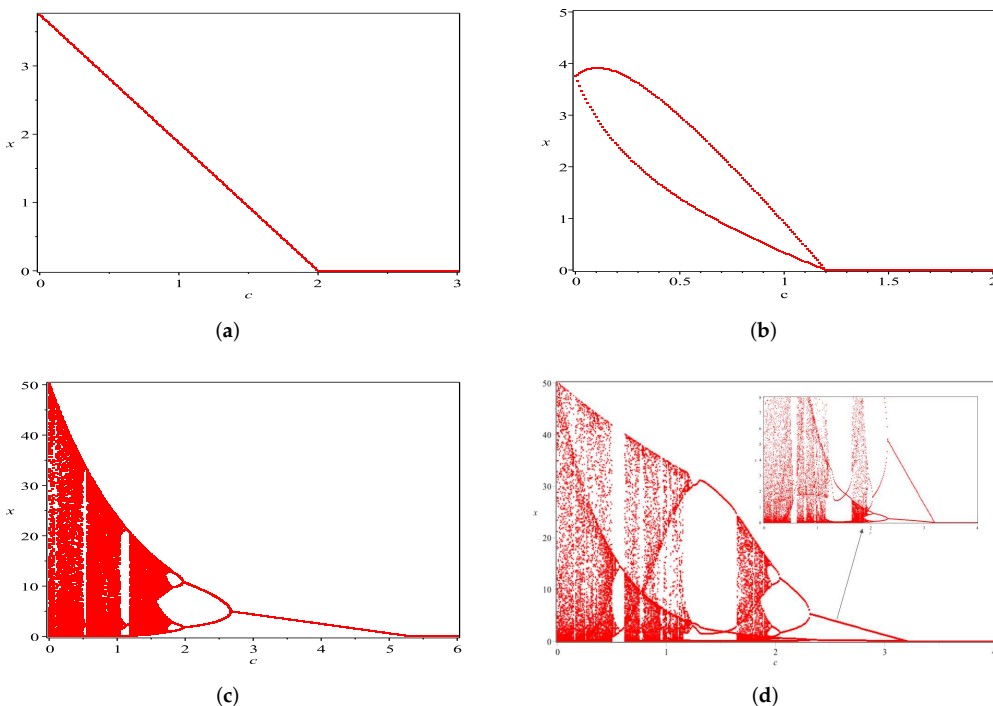

**Figure 2.** Bifurcation diagrams of system (3). **(a)** $\alpha = 1.5, \beta = 0.4, \gamma = 1.5, \delta = 2, 0 < c < 3$; **(b)** $\alpha = 1.5, \beta = 0.4, \gamma = 2.5, \delta = 2, 0 < c < 2$; **(c)** $\alpha = 4, \beta = 0.4, \gamma = 1.5, \delta = 2, 0 < c < 6$; **(d)** $\alpha = 4, \beta = 0.4, \gamma = 2.5, \delta = 2, 0 < c < 4$.

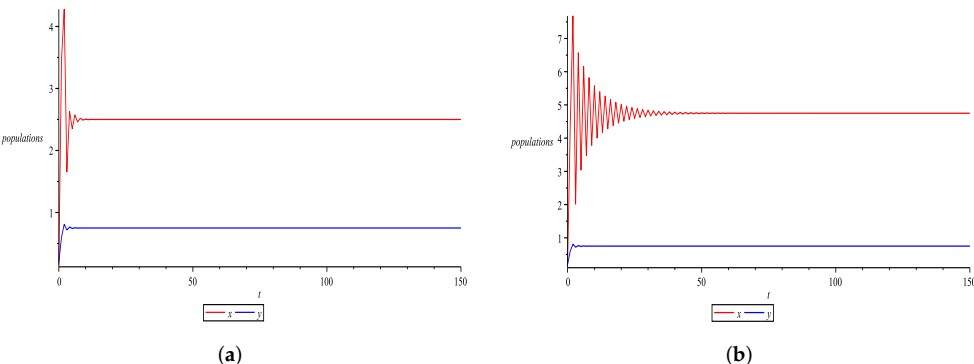

**Figure 3.** The stability of interior fixed point of systems (3) and (4), respectively. **(a)** $k = 0$; **(b)** $k = 0.3$.

**Example 2.** *Next, taking $k = 0.5$ with initial value $(x_0, y_0) = (0.15, 0.2)$ and other parameter values of system (4) are as follows:*

*(e)   Varying c in range $-4 < c < 4$, and fixing $\alpha = 1.5, \beta = 0.4, \gamma = 1.5, \delta = 2$;*
*(f)   Varying c in range $-4 < c < 4$, and fixing $\alpha = 1.5, \beta = 0.4, \gamma = 2.5, \delta = 2$;*
*(g)   Varying c in range $0 < c < 8$, and fixing $\alpha = 4, \beta = 0.4, \gamma = 1.5, \delta = 2$;*
*(h)   Varying c in range $0 < c < 8$, and fixing $\alpha = 4, \beta = 0.4, \gamma = 2.5, \delta = 2$.*

*For case (e), from Theorem 12, one can see that system (4) has no positive fixed point if $c > 4$, conversely, if $0 < c < 4$ is true, $E^{**}(3.75 - 0.9375c, 0.75)$ is a sink. The image is depicted in Figure 4a. For case (f), when $0 < c < 2.4$, system (4) has a unique positive fixed point $E^{**}(3.75 - 1.5625c, 1.25)$, but it is unstable. When $c > 1.2$, there is no positive fixed point. Refer to Figure 4b for details. For case (g), there exists flip bifurcation at $E^{**}(5, 0.75)$ around $c^* = 5.333$. Furthermore, $\varpi_1 = 1.25, \varpi_2 = 4.4741 > 0$ can be easily calculated, then we can infer that the period-2 point is attracting by Theorem 19. The flip bifurcation diagram is shown in Figure 4c. For case (h), when $c > 6.4$, system (4) has no positive fixed point. If $0 < c < 6.4$ exists,*

$(x^{**}, \frac{\gamma}{\delta}) = (10 - 1.5625c, 1.25)$ *can be easily calculated while it is unstable. The image is shown in Figure 4d.*

*Consistent with the results of Example 1, this example again verifies the effect of the internal growth rate of population y on the stability of system (4). Moreover, a significant finding can be observed: cover can increase the densities of the first species x, and thus reduce the chance of the extinction of the first species x. Note that in Figure 4a,b, the dynamic behavior is also very complex, which is worthy of further exploration in the future (c < 0 means that y has a favorable effect on x).*

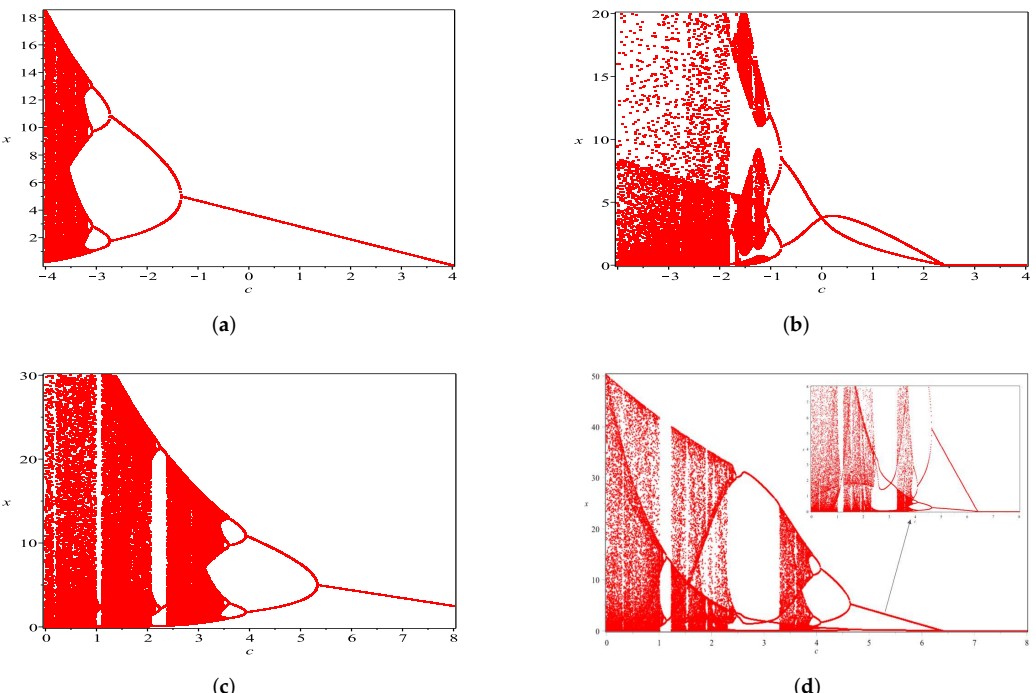

**Figure 4.** Bifurcation diagrams of system (4). (**a**) $(\alpha, \beta, \gamma, \delta) = (1.5, 0.4, 1.5, 2), -4 < c < 4$; (**b**) $(\alpha, \beta, \gamma, \delta) = (1.5, 0.4, 2.5, 2), -4 < c < 4$; (**c**) $(\alpha, \beta, \gamma, \delta) = (4, 0.4, 1.5, 2), 0 < c < 8$; (**d**) $(\alpha, \beta, \gamma, \delta) = (4, 0.4, 2.5, 2), 0 < c < 8$.

**Example 3.** *To better illustrate the role of cover on the first species, we chose k as the bifurcation parameter with initial value $x_0 = 0.15, y_0 = 0.2$. We consider the following two cases.*

*(i)   Fixing the parameters $(\alpha, \beta, c, \gamma, \delta) = (4, 0.4, 4, 1.5, 2)$;*
*(j)   Fixing the parameters $(\alpha, \beta, c, \gamma, \delta) = (4, 0.4, 4, 2.5, 2)$.*

*This example again validates the findings of the previous two examples. It is worth mentioning that when other conditions remain unchanged, the refuge is too large, which is not conducive to the stability of the population from Figure 5a,b. Furthermore, system (4) at the positive fixed point $E^{**}(x^{**}, \frac{\gamma}{\delta})$ experiences flip bifurcation. Figure 5 also shows periodic-2, 4, 8 orbits. The maximum Lyapunov exponent is shown in Figure 5c.*

**Example 4.** *For the parametric values $(\alpha, \beta, \gamma, c, \delta, k) = (4, 0.4, 4, 1.5, 2, 0.8)$ with initial value $x_0 = 0.15, y_0 = 0.2$, the controlled system (32) can be written as*

$$\begin{cases} x_{n+1} = \rho x_n \exp\left(4 - 0.4x_n - 4(1 - 0.8)y_n\right) + (1 - \rho)x_n, \\ y_{n+1} = \rho y_n \exp\left(1.5 - 2y_n\right) + (1 - \rho)y_n, \end{cases} \tag{34}$$

*then controlled system (34) has unique positive steady-state $(x^{**}, y^*) = (8.5, 0.75)$. Based on Theorem 20, $E^{**}$ is locally asymptotically stable if $0 < \rho \leq 0.58$. Moreover, $E^{**}$ is unstable when $\rho \in [0.59, 1)$. Figure 6 agrees with this, where $\rho = 0.58$ in Figure 6a and $\rho = 0.59$ in Figure 6b.*

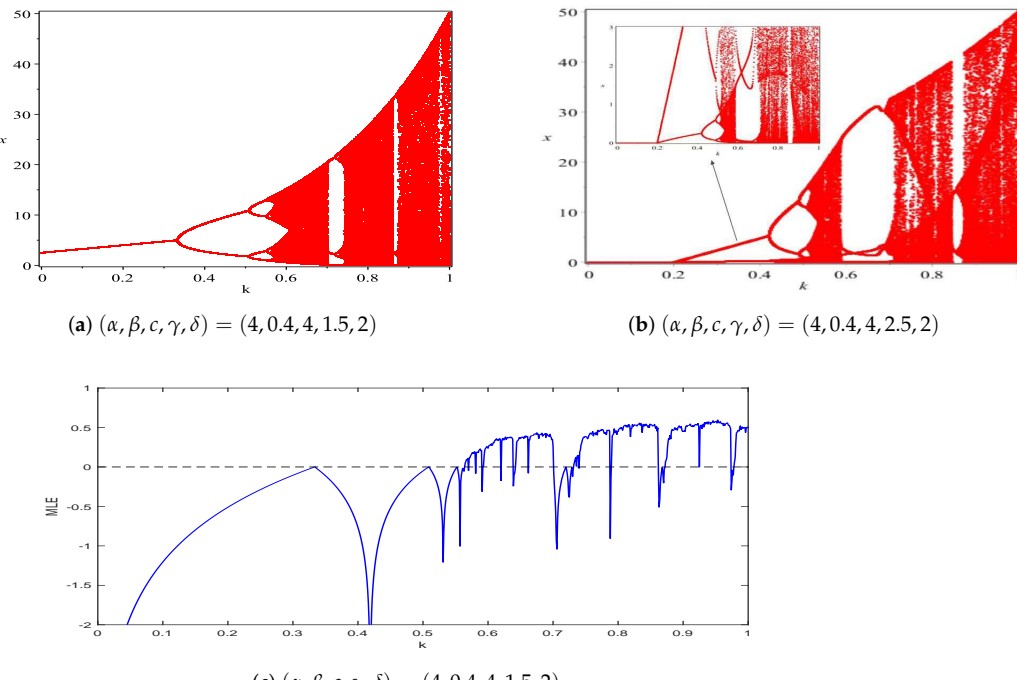

**(a)** $(\alpha, \beta, c, \gamma, \delta) = (4, 0.4, 4, 1.5, 2)$  **(b)** $(\alpha, \beta, c, \gamma, \delta) = (4, 0.4, 4, 2.5, 2)$

**(c)** $(\alpha, \beta, c, \gamma, \delta) = (4, 0.4, 4, 1.5, 2)$

**Figure 5.** (**a**,**b**) Bifurcation diagrams of system (4); (**c**) the maximum Lyapunov exponent of the positive fixed point of system (4).

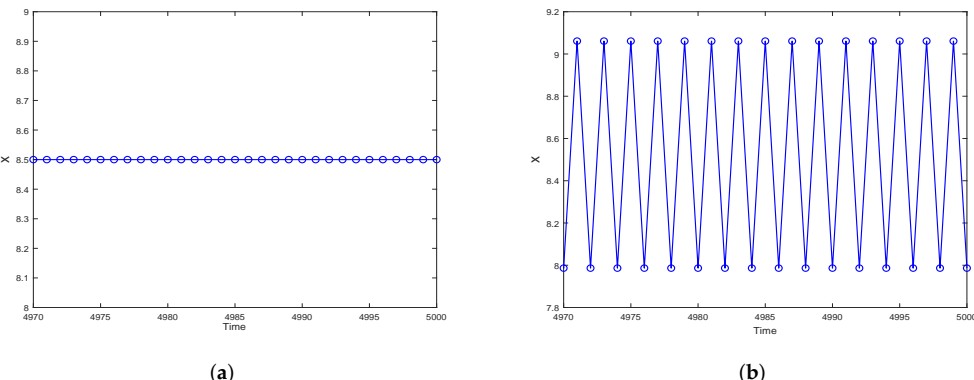

**(a)**  **(b)**

**Figure 6.** Time series of the first species for the controlled system (32) with different $\rho$. (**a**) $\rho = 0.58$; (**b**) $\rho = 0.59$.

## 5. Summary and Discussion

During the past two decades, amensalism systems have been investigated by several mathematicians, ecologists, and biologists due to their applications to biomathematics. However, discrete amensalism systems have not yet received sufficient attention from researchers. In this paper, we proposed a discrete amensalism system with a cover for the first species.

Firstly, we obtained the discrete model by using the method of piecewise constant argument. The dynamic behavior of discrete-time amensalism systems (3) and (4) are analyzed in detail. It is proved that the two discrete models considered have the same fixed points as their corresponding continuous models. However, the dynamic behaviors of systems (2) and (4) are quite different.

(1) In system (2), Theorem 1 (2) in the Section 1 shows that if the positive equilibrium exists, it is globally stable. This means for any positive initial condition, the solution

will eventually approach this equilibrium. However, for the discrete system (4), noting that $\exp\{x - 1\} > x$ for $x > 0$ always holds, hence

$$\alpha > c(1 - k)M = c(1 - k)\frac{\exp\{\gamma - 1\}}{\delta} > c(1 - k)\frac{\gamma}{\delta}. \tag{35}$$

That is, under more restricted conditions than that of Theorem 1 (2), we could only obtain the permanence result (see Theorem 16).

(2)　Since system (4) allows only one positive equilibrium, and under more restricted conditions we could only obtain the permanence result, it is natural and important to find out the conditions which guarantee the global attractivity of positive equilibrium. By developing the analysis technique of Chen [38] and Li and Chen [39], we finally obtained a set of sufficient conditions for the global attractivity of the positive equilibrium (see Theorem 18). The condition seems to be the best one, since for single species discrete model

$$y_{n+1} = y_n \exp\{\gamma - \delta y_n\}. \tag{36}$$

$0 < \gamma < 2$ is the best condition to ensure the global attractivity of the positive equilibrium, and with the increasing of $\gamma$, the system may have a $2, 4, 8, \ldots$ period solution, and finally leads to chaos.

(3)　Systems (3) and (4) have three boundary fixed points and at most one interior fixed point. The topological types of their fixed points are completely classified. It seems that the local stability property of the equilibria becomes complicated. There are three cases about the stability of $E_2$ and $E^{**}$. Here, the topological types of $E_1$, $E_2$, and $E^{**}(x^{**}, \frac{\gamma}{\delta})$ can be found in Theorems 4, 13, and 14, respectively. Moreover, compared with the system (2), we confirm that system (4) experiences flip bifurcation at two boundary fixed point $E_1(\frac{\alpha}{\beta}, 0)$, $E_2(0, \frac{\gamma}{\delta})$ and the positive fixed point $E^{**}(x^{**}, \frac{\gamma}{\delta})$ separately.

The above results show that we have good reasons to believe that the dynamic behavior of the discrete-time model is richer than that of the continuous-time model. In the end, a hybrid control strategy is implemented to control the flip bifurcation. The theoretical results of this paper are supported by numerical simulation, which also indicates some exciting outcomes:

(I)　We conclude that, for some fixed parameter values, the intrinsic growth rate of the second population plays a major role in the stable coexistence of two species, which is supported by numerical simulations in Examples 1 and 2. This is a novel finding compared with the previous research results [22].

(II)　Based on Theorem 14 of this paper, one can deduce that $0 < \alpha \leq 2, 1 - \frac{\alpha\delta}{c\gamma} < k < 1, 0 < \gamma < 2$ or $\alpha > 2, 1 - \frac{\alpha\delta}{c\gamma} < k < 1 - \frac{(\alpha-2)\delta}{c\gamma}, 0 < \gamma < 2$ holds, two species reach stable coexistence in the system (4). This conclusion is different from the results of Xie, Chen, and He [22].

(III)　With the change of cover intensity of the first population, system (4) experienced interesting and complex dynamic characteristics, including population stable coexistence, multiple invariant closed orbits in different chaotic regions, and the onset of chaos suddenly. According to Figure 5, one can observe that the $k$ value is small, it is conducive to the stability of the first population. However, it may destabilize the first population causing more complex dynamical behaviors when the $k$ value exceeds a certain threshold.

At the end of the paper, we would like to mention that one of the reviewers argued that "I am not sure that it has been observed in that kind of systems, in nature". Though the phenomenon of amensalism is very common in nature, it seems that biologists have done little work in this direction, and it is really a difficult thing for us to find out suitable examples in nature. However, we found from Baidu Baike that "Although Brazil nut tree is tall, it also has the object of being defeated—strangled banyan. The seeds of banyan tree first settled on the branches of Brazil nut tree, and then grew up day by day. Vines

that strangled banyan slowly wrapped around Brazil nut tree while absorbing nutrients. After several years, the nut tree was only an empty shell." (https://baike.baidu.com/item/%E5%B7%B4%E8%A5%BF%E6%A0%97/6802281?qq-pf-to=pcqq.c2c, accessed on 22 July 2022). In other words, banyan trees are harmful to Brazil nut trees. We believe that the relationship between these two trees is amensalism, since the Brazil nut tree is extremely useful to human beings, and humans will naturally deal with banyan trees, which can limit the impact of banyan trees to a controllable range. We consider this may be a possible suitable example, however, we need some more detailed data to support our conjecture. At present, we fail to do so.

**Author Contributions:** Q.Z., F.C. and S.L. contributed equally to the writing of this paper. All authors have read and agreed to the published version of the manuscript.

**Funding:** This work was supported by the Natural Science Foundation of Fujian Province (2020J01499).

**Data Availability Statement:** Not applicable.

**Acknowledgments:** The authors would like to thank three anonymous reviewers for their valuable comments, which greatly improved the final expression of the paper.

**Conflicts of Interest:** The authors declare no conflict of interest.

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
