# Peer review of "Complex Dynamics Analysis of a Discrete Amensalism System with a Cover for the First Species"

_axioms, doi:10.3390/axioms11080365_

Round 1

Reviewer 1 Report

COMPLEX DYNAMICS ANALYSIS OF A DISCRETE AMENSALISM SYSTEM WITH A COVER FOR THE FIRST SPECIES is a very interesting manuscript that deals with a subject of crucial importance, and which attracts a lot of attention from researchers, in the field of Biomathematics.

After a very complete and enlightening introduction to the problem, the authors justify very well, both from the point of view of applications to reality and from a mathematical point of view, the choices of the model to be studied.

The study is presented in a very clear and rigorous way and, above all, with an impeccable order.

The results are verified and illustrated through carefully chosen numerical experiments.

The authors adequately emphasize their contribution, and respective importance, to the topic.

In my opinion, the manuscript can be published as is.

Author Response

Dear Sir, 

   Thank you for your valuable comments.  However, the other two reviewers have different opnion, and they think it is better for us to compare the result with continous one and to give the suitable example in nature.

    Hence, we revise the introduction section, state  some possible ammensalism example, and cited the results of Xie, Chen and He.  We also revised the conclusion section to compare our results with the results of Xie, Chen and He.  In the couclusion section, we also give a possible application, the relationship of banyan tree  and Brazil nut tree. 

      To support our declaration, we add several new references. 

    Hope this time everything are O.K. 

Reviewer 2 Report

The paper is devoted to the analysis of the dynamics of two species populations with amensalism relation, i.e. the first species is suppressed by the second while the second is not affected by the first. This is well known problem for continuous problem statement. The authors study its discrete version, motivated by its better suitability to the species with non-overlapping generations. This is novel. One aspect of this model is elaborated deeper. First species may have a refuge of given 'size' from the suppression of their cohabitants. The influence of the refuge size to the system dynamics is studied, also for discrete statement. The study is done both in theory and by numerical experiments.

The paper structure is logical and comprehensible, the mathematical level is high, the results are presented fully and clearly.

To my mind, there are the following two drawbacks in the paper.

1. The authors claim the behavior of discrete systems is more complex then that of continuous counterparts. I recommend to add more detailed discussion on this matter. Where it is more complex? Does discrete system have more fixed points? Does it have more topological types for fixed points? Does it have more complex bifurcation pattern? Of course, the results for continuous case can be found in appropriate literature, but it is advisable to give the reader some comparison and conclusions immediately in the paper.

2. The authors claim discrete approach is more suitable for non-overlapping generations. This is in fact, an 'a priori knowledge', based on previous study. Then the authors perform the study and obtain certain results of their own. Do these results really support the prior claim? In other words, does authors' discrete model fit the case better than continuous one? Some comparison with continuous model might be presented, as in item 1.

Minor remark. MLE abbreviation at page 26 should be disclosed.

Author Response

Dear Sir, 

  Thank you for your valuable comments.

  1. Comment: The authors claim the behavior of discrete systems is more complex then that of continuous counterparts. I recommend to add more detailed discussion on this matter. Where it is more complex? Does discrete system have more fixed points? Does it have more topological types for fixed points? Does it have more complex bifurcation pattern? Of course, the results for continuous case can be found in appropriate literature, but it is advisable to give the reader some comparison and conclusions immediately in the paper.   
  2. Reply: Thank you for your comment, in the revised version of our paper, we state the main results of Xie, Chen and He in the introduction section.  We then compare our results with the corresponding results of Xie, Chen and He in the conclusion section, with those comparison, it seems that our study becomes meaningful. Thank you again, we think it really a good method in thinking and studying. We will try to absorb this idea in our future study. 

2. Comment: The authors claim discrete approach is more suitable for non-overlapping generations. This is in fact, an 'a priori knowledge', based on previous study. Then the authors perform the study and obtain certain results of their own. Do these results really support the prior claim? In other words, does authors' discrete model fit the case better than continuous one? Some comparison with continuous model might be presented, as in item 1.

Reply:   Thanks again for point out this, yes, " discrete approach is more suitable for non-overlapping generations. This is in fact, an 'a priori knowledge', based on previous study."     When we declare this, it really comes from other paper, we had no idea on the difference of our system and the continous one, now, to answer this,  we  try to give a clearly answer, so we give detail comparison in the couclusion, and as you can see, yes , the dynamic behaviors of the system becomes very complicated.   

    Thank you again for your comments, which lead us to reflect our study and try to discribe our finding clearly. 

    We learned many things from your comments.  You are a excellent researcher.  

    Hope this time everything are O.K. 

Reviewer 3 Report

  Amensalism is admittedly a common and interesting phenomenon. The mathematics underlying it, is rather simple. Still as this paper confirms it can contain a lot of dynamical structure, including chaos. That was known before and I am not sure that it has ever been observed in that kind of systems, in nature.

  This paper is about the effect of cover. It has obviously an effect in the math, as the value of “k” enters in the bifurcation analysis. I would have been interested in examples where the effect of cover was observed in nature…

   It remains that however somewhat dry, the mathematical analysis is solid and I cannot find any good reason to object for its publication, as is.

Author Response

Dear Sir

    Thank you for your comment,  we try to find out suitable example in nature, however, it really not an easy thing.

     After those days search literature, we could find out a possible suitable example: banyan tree  and  Brazil nut tree, we wrote this in the final paragraph of our paper. 

     Anyway, it really a difficult thing for us to overcome the difficulty  between theoretical research and practical biological background. 

    Hope our revise could in some sense answer your comments.